# Exploring the impact of health literacy on pregnant women from ethnic minority groups: A scoping review

Sarah E. Feldman[1], Laura Lennox[1,2], Natasha Dsouza[1,2], Keivan Armani [1,2,3]*

1 Imperial School of Public Health, Imperial College, London, United Kingdom, 2 Department of Primary Care and Public Health, NIHR ARC NWL, School of Public Health, Imperial College, London, United Kingdom, 3 Faculty of Pharmaceutical Sciences, UCSI University, Kuala Lumpur, Malaysia

* karmani@imperial.co.uk

## Abstract

### Objective

Health Inequalities refer to disparities in healthcare access and outcomes based on social determinants of health. These inequalities disproportionately affect Black, Asian, Minority Ethnic (BAME) groups, particularly pregnant women, who face increased risks and limited access to care due to low health literacy. Maternal mortality rates for BAME women can be up to four times higher than for white women. This scoping review aimed to assess the impact of health literacy on BAME pregnant women's health outcomes and experiences. Objectives included evaluating health literacy improvement tools, engaging stakeholders through co-production, and identifying persisting health inequalities.

### Data sources

A scoping review using the Arksey and O'Malley's framework was conducted. A specific search strategy was developed with a research librarian across three databases: EMBASE, Medline, and the Maternity and Infant Care (MIC) database. Patient, Public Involvement, and Engagement (PPIE) members were consulted from the outset to co-design the research question and to provide feedback on the findings.

### Study appraisal and synthesis methods

Out of 1958 articles, 19 were included in the study, with 47% from the US and 21% from Denmark. Articles were published from EMBASE. 47% of the articles measured health literacy, while 53% implemented health literacy interventions, such as digital or community-based approaches.

### Results

All 19 articles highlighted lower health literacy in BAME women compared to other groups. Ten proposed recommendations, while others emphasized the impact of social determinants of health, collectively underscoring the need for more research on BAME health.

**Data Availability Statement:** All relevant data are within the manuscript and its Supporting Information files.

**Funding:** The author(s) received no specific funding for this work.

**Competing interests:** The authors have declared that no competing interests exist.

## Conclusions

The review underscores the inadequate health literacy and patient experience of BAME pregnant women. It also highlights the potential of digital health interventions to improve health literacy and health outcomes. The findings call for increased research into health literacy tools for BAME pregnant women. Healthcare systems, including the NHS, should allocate resources to enhance digital health interventions and address health inequalities in BAME groups during pregnancy.

## Introduction

Maternal health and wellbeing are classified as the health of a person throughout pregnancy, childbirth, and throughout the postnatal period [1]. There are different ways to measure maternal health, focusing on both the mother and the new-born baby. For the purposes of this review, we focused on maternal health outcomes which are measured in different ways but focus more on the prevalence of adverse health outcomes in pregnancy, childbirth and the postnatal period [2]. BAME is a collective acronym that stands for Black, Asian and Minority Ethnic in the United Kingdom [3]. Although we have used the term BAME throughout this manuscript, we have been cognisant of the fact that ethnic minority groups are labelled differently in different countries, for example, Minoritized Ethnic Groups in the United States. BAME communities are more likely to have adverse health outcomes for a myriad of reasons stemming from systemic to individual causes [4]. Though there has been documented evidence for such health inequalities, the COVID-19 pandemic revealed the size and depth of the issue more clearly than before [5]. BAME groups suffering from adverse health outcomes could be construed as the direct consequence of social constructs. The purpose of health literacy is to enable patients/members of the public to make informed decisions about their health [6]. The concept of health literacy is directly linked to socioeconomic status and thus connected to the social determinants of health [6, 7].

Health inequalities affect maternal health among ethnic minority groups, including BAME communities, in every society globally. BAME women are more likely to die in childbirth than white women; Asian women are two times more likely, while Black women are up to four times more likely to die from childbirth complications [8, 9]. The international push towards decreasing maternal mortality is only possible if health inequalities are addressed and at risk BAME groups are given the proper attention during childbirth experiences [10].

There have been a series of reports published on the increasing maternal mortality rates of ethnic minority women globally, as seen in the articles found within this scoping review. Although the majority of the literature comes from Global North, there were a couple of studies from Global South [8, 9, 11–13], which on its own seems to be highly debatable and could highlight the lack of data in the peer-reviewed academic literature [13, 14]. Joseph et al. highlights the main cause of maternal death in the US lay at the hands of the provider, particularly in maternal mortality surveillance [15] A scoping review from Garcia et al., found that from over 2,000 articles, only five were deemed suitable in discussing antenatal healthcare interventions for Black women in the UK [16].

From the choice to seek care, to the interactions with providers, and finally to the point of delivery, adverse experiences affect not only the patient but their caregivers and their families [17, 18]. Multiple studies have found positive patient outcomes from positive patient

experiences [19, 20]. This is extremely important during pregnancy and childbirth, as interactions with caregivers, clinicians, and medical support staff are increased. Maternal health outcomes are linked to the care given to both the mother and any birthing support partner. The connection between health outcomes and patient experience should not be overlooked in literature, as the improvement of patient experience in healthcare settings could provide improvement in overall health.

## Objectives

While many studies have begun investigating the importance of health literacy and the impact of health inequalities within BAME groups, there is limited knowledge of health literacy within the context of maternal health outcomes [18, 19]. Until there are significant studies produced where BAME women are the focus, there will be a continued gap of knowledge in this area of research.

This review aimed to explore the current literature on the impact of health literacy on the maternal health outcomes and patient experience of BAME pregnant women. The other aims were to evaluate potential interventions for increasing health literacy and self-advocacy in BAME populations, whilst identifying factors that continue to harm BAME pregnant women and to explore the public health implications from those interventions.

## Methods

The six-stage Arksey and O'Malley framework [21] was used to document the methods of review, as proposed by the Joanna Briggs Institute Reviewer's Manual [22]. The purpose of this framework is to systematically lay out the design and the presentation of the findings in the following six stages i.e., 1) Identifying the review question, 2) Search for relevant studies, 3) Study selection and eligibility, 4) Charting the data, 5) Collating, summarising, and reporting the results, 6) Consulting stakeholders.

Through an iterative process the authors worked in collaboration with the patient representatives to identify a couple of suitable questions in the realm of patient empowerment tools and health literacy, for this scoping review. \We chose the Patient/Public, Concept, and Context (PCC) framework to formulate and finalise the research question. As our research relied on the lived experiences of patients/public, we found the PCC framework to be the most appropriate framework to properly acknowledge both the context of maternal health outcomes and the general concept of health literacy. Based on discussion, the agreed upon research question was proposed as "How does health literacy of BAME pregnant women impact maternal health outcomes and patient experience?" A secondary research question that was proposed was "what interventions to impact health literacy in BAME pregnant women have been used/ employed and what is their impact on health outcomes and patient experience?" Both questions were used as a guide for the literature found in the scoping review.

## Eligibility criteria

The search for studies was carried out in July 2023 using three databases: Medline, Embase, and Maternity and Infant Care Database (MIC). Terms involving pregnancy AND health literacy AND BAME or ethnic minority group were combined for maximum results, see S1 Appendix for detailed search terms that were used. To further enhance the search, the Prospero systematic review database was used as a guide. By searching for articles that had corresponding key words to the study, we checked systematic reviews to see similarities and differences in already published works. The final search was carried out on July 27th, 2023. The final search was uploaded to Covidence where duplicates were removed. Covidence is an

online screening tool to streamline the systemic review process, where researchers can scan abstracts and titles of articles, conduct full text screening, and manage all sources in one location. Covidence was also used to create the PRISMA flow chart, as seen in the results.

## Study selection

Titles and abstracts were screened by one of the authors (SEF) before full text screening. Conflicts were discussed with the supervisory team (KA & LL) and agreed upon before moving to the next stage of data extraction. Studies published before 2011 were excluded. The Race and Health Observatory (RHO), established in 2021, published a report on ethnic inequalities in healthcare within the NHS [24]. The data collected and published in the report spans from 2011–2021 and is the most recent and up to date evidence of the impact of race within the healthcare system. Thus, the dates of the report were used as a guide for the dates chosen within the review. Randomised Control Trials (RCTs), qualitative or mixed method studies were included. Studies that included either physical health literacy interventions or abstract interventions were also included. The rationale for these inclusions and exclusions was to create a cohesive finding of studies focusing on patient outcomes and health literacy, which meant including studies that focused on maternal welfare from patient perspectives. Studies excluded during the full-text screening stage were put into the categories described in Table 1.

## Data extraction

A data extraction table was designed to highlight the key components from the studies that were chosen. Data for the author, year published, title, country of study, number of participants, study design, aims/objectives, approach to assessing health literacy, method of delivery, and method of data collection were extracted and categorised into the excel spreadsheet. Since each study did not always have an intervention, the general method of observing health literacy was recorded. The differences in the results were discussed among the researchers and categorised as either having a physical intervention (digital or community-based) or a survey-based observation.

Different categories of findings, along with gaps, were identified throughout the studies. Categories with similar results, were the first to be compared, as the outcome of the study was particularly interested in this category. Differences in health literacy interventions categorised as digital interventions, community-based interventions or other. The effectiveness of the interventions was also noted by extracting the data from the outcome of the study. Results were calculated using simple descriptive statistics along with narrative summaries to explain the outcomes and implications of each study.

**Table 1. Inclusion and exclusion criteria.**

| Criterion | Included | Excluded |
|---|---|---|
| Time period | 2013–2023 | Dates outside this range |
| Language | English | Any other language |
| Study Design | RCT, qualitative, quantitative, mixed methods, cohort study | Systematic reviews, narrative reviews, scoping reviews, poster presentations |
| Study Focus | Health literacy and or BAME populations | Vaccinations, breastfeeding, pre-existing conditions |
| Population | Pregnant Women/Post-partum (42 days after birth) | non-pregnant women of reproductive age |
| Geographic Location | Any | |

## Data synthesis

Throughout the process of the review, we consulted stakeholders through discussions and feedback session with the Patient and Public Involvement and Engagement (PPIE) group for the Innovation and Evaluation Theme within the Applied Research Collaboration for North-west London based at Imperial College London. Primary findings from the eligible studies were presented to the PPIE group. The purpose of this meeting was to discuss the results that were found with stakeholders and to gain feedback on possible areas of further discussion. Though this is not a requirement for a scoping review, consulting the stakeholders was deemed necessary to gain feedback on the topic, its findings, the implication of the results as well as the accessibility of the language/terminologies used to describe the findings. The qualitative content analysis of the text in our manuscript was "descriptive" in nature. We did not apply content/thematic analysis techniques, as these techniques are not prescribed for scoping reviews. We acknowledge that JBI's meta-aggressive methos were beyond the scope of this review [48].

## Results

### Study selection

The scoping review identified 2028 possible sources for inclusion. Through Covidence 70 duplicates were removed, and another 12 were removed through manual screening, leaving 1958 articles to screen. After title and abstract screening, the number of texts decreased to 164. Through full text screening, 145 articles were excluded: (89 = wrong study set up, design, or population, 56 = irrelevant characteristics). In total, 19 studies were identified as suitable and used for data extraction, as detailed in the PRISMA flowchart in Fig 1. The 19 full texts were then fully analysed and relevant data and study characteristics were extracted from each article.

### Study characteristics

Key study characteristics are described in full and presented in tabular format in Table 2. The geographical distribution of 19 articles is diverse, spanning 8 countries, with significant contributions from the United States (47%) (n = 9) and Denmark (21%) (n = 4). Notably, only two articles originated from the Global South, specifically Thailand and Brazil, representing middle/high-middle-income countries (Fig 2) [11, 12].

The observed study designs are varied, predominantly featuring qualitative methodologies (42%) (n = 8) and mixed methods (32%) (n = 6). Cross-sectional studies constitute 21% (n = 4), and a single randomized control trial (RCT) by Damsted Rasmussen et al. comprises 5% (n = 1) [23]. The participant sample sizes varied across all studies amounting to 11,235, with individual sample sizes ranging from 11 [24] to the largest study at 4,150 participants [23].

This review examined data collection methods, mainly focusing on patient experience, health literacy, and attitudes towards practice. Most studies (79%) (n = 9) utilised interviews, questionnaires, surveys, or a combination (26%) (n = 5). The remaining studies (21%) (n = 3) employed methods such as focus groups or specific training, demonstrated by the MAMAACT system that was implemented for both patients and midwives, comparing health outcomes before and after implementation [23, 25]. Out of the 19 articles, 37% (n = 12) were published between 2015 and 2018, while the remaining 63% (n = 7) were published between 2019 and 2023.

Between studies chosen for the scoping review, there are differences in which BAME groups are represented in the patient population influenced by both the publication countries and the

Health Literacy in Pregnant BAME Women

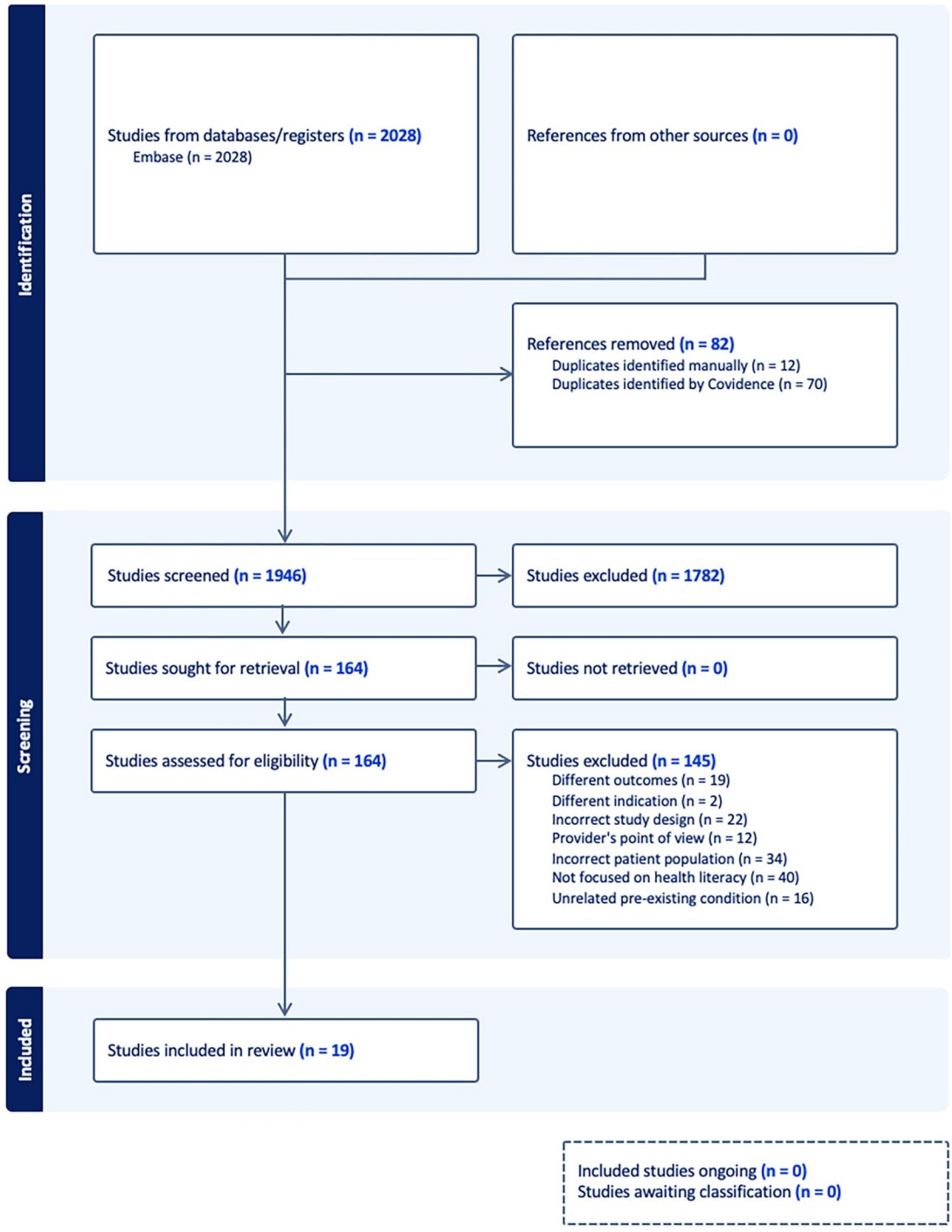

**Fig 1. PRISMA flow diagram.**

specific ethnic minorities considered. Four of the studies (21%) (n = 4), all from the US, looked at Black women's experience with health literacy during pregnancy or their experience with childbirth in general. Of these, three studies focused on African American women, one from

**Table 2. Chosen sources for data extraction.**

| Author, Date and country | Aims/ Objectives | Sample size | Study Design | Intervention (if applicable) | Outcome(s) | Findings and Recommendations | Considerations |
|---|---|---|---|---|---|---|---|
| Alio et al. (2022), USA | To establish a community consortium charged with the development of a research agenda to address racial inequities leading to SMM/ MM in the greater Rochester area. | 44 | Qualitative study | Consortium-Community based intervention | Gained knowledge on the importance of quality patient/provider interaction along with increasing health literacy in all different communities | Further interventions to improve health literacy should then include women of all economic backgrounds. | Challenge of small samples size, room for more diversity, selection bias (chosen participants had negative experiences with childbirth) |
| Bains et al. (2021), Norway | Explored factors associated with newly arrived migrants' understanding of information provided by maternity staff. Also determined which maternal health topics the women had received insufficient coverage of. | 401 | mixed methods study | Migrant Friendly Maternity Care Questionnaire-Community health Intervention | Found suboptimal provision of interpreting services. Suggest that current policies are yet to be put into consistent practice. | Targeted interventions should be applied to adapt healthcare services to linguistically diverse patients, including the provision of tailored health education and prenatal classes that consider the specific needs of newly arrived migrants. | Postpartum care is fragmented in Norway, social desirability bias, caution must be taken when generalizing the results |
| Barnett et al. (2022), USA | To elicit information related to barriers, successes, and existing opportunities to provide equitable care and services to families during pregnancy, labour and post-partum | 31 | Qualitative study | N/A | Identification of 7 themes experienced by Black women who give birth in the U. S | Informed the strategic plan that will guide the work of CelebrateOne to reduce infant mortality in the next 5 years and beyond | Challenge of small sample size and unable to generalize the data to expand across the country. |
| Blackwell et al. (2020), USA | To test a maternal health education intervention (T4B) to see if it improves access to prenatal health care and information, improves prenatal health-seeking behaviours in pregnant women, in central Brooklyn | 58 | Mixed methods study | T4B mHealth texting program-Digital Intervention | Feedback of the prenatal health electronic messages through texting had a positive and was highly compatible to provide pregnant women in central Brooklyn with information. | Post-test 4 weeks after baseline test administered to participants showed potential for increased used. | Needed a larger sample size to realistically measure the impact of the app's benefits along with other digital health interventions |
| Brorsen et al. (2021), Denmark | To investigate potential ethnic differences in Active engagement with healthcare providers during pregnancy using a phone-based survey that was conducted in the six most spoken languages among pregnant women in Denmark. | 1898 | A cross-sectional study | N/A | Showed that European, African and Asian immigrant women reported lower Active engagement with healthcare providers than did ethnic Danish women. | Previous research suggests that service provision is better centred around users' needs if healthcare providers and users share the same health understanding and have similar cultural backgrounds | Study did not accurately represent the Danish education system as well as differences within socioeconomic statuses |

(*Continued*)

**Table 2.** (Continued)

| Author, Date and country | Aims/ Objectives | Sample size | Study Design | Intervention (if applicable) | Outcome(s) | Findings and Recommendations | Considerations |
|---|---|---|---|---|---|---|---|
| Damsted Rasmussen et al. (2023), Denmark | To analyse effectiveness of the MAMAACT intervention in improving women's health literacy of active engagement with healthcare providers, healthcare system navigation, and complication management. | 4150 | RCT | MAMAACT training- Community based intervention | The intervention did not demonstrate an improvement in pregnant women's health literacy in Active engagement with healthcare providers and navigating the healthcare system | Combined use of digital and printed materials might be beneficial to a diverse group of pregnant women, when implemented together with the training of maternity care providers. | System-related barriers in antenatal care seem to be the main explanation for the lack of increase in health literacy in the intervention group. |
| Dayyani et al. (2018), Denmark | To explore how non-Western ethnic minority pregnant women with GDM in Denmark experience the hospital-based information about GDM and how they integrate this information into their everyday life. The secondary aim was to investigate the role played by health literacy and distributed health literacy. | 11 | qualitative study | N/A | Showed that lack of individually based instruction left some women less understanding or misunderstanding of important issues. | future studies investigating differences between women with previous GDM/ first-time GDM could provide valuable information for clinicians. | The focus on the health literacy perspective may have limited a more thorough phenomenological description of the women. |
| Jennifer Foster et al. (2015), USA | to encourage low resource African American women in the U.S., who live in neighbourhoods with the highest rates of adverse pregnancy outcomes and infant mortality in Atlanta, Georgia, to adhere to medically recommended prenatal vitamins, prenatal clinic visits. | 14 | Qualitative study | mHeath text messaging service- Digital Health intervention | Found the messages helpful. wanted more focus on mental health and requested reminders for their babies' well-child appointments | All stakeholders contributed to a better understanding of a mHealth application that would potentially be user-friendly and effective with the population under study. | Technological difficulties/computer science delays, phone plans make the study not fully inclusive |
| Gilder et al. (2019), Thailand | To analyse the prevalence and predictors of low Health literacy, and focus group discussions to gather qualitative data from women about proposed and actual posters used in the campaign. | 525 | mixed methods study | N/A | showed the importance of piloting posters before widescale implementation and the significant challenges facing health workers seeking to broadly distribute health information in marginalized communities with low educational levels | Verbal communication remains a key method of messaging with individuals of low health literacy and educational system strengthening and audio-visual messaging are critical for improvement of health outcomes. | Staff shortages and multiple languages in the study were challenges |

*(Continued)*

**Table 2.** (Continued)

| Author, Date and country | Aims/ Objectives | Sample size | Study Design | Intervention (if applicable) | Outcome(s) | Findings and Recommendations | Considerations |
|---|---|---|---|---|---|---|---|
| Guendelman et al. (2017), USA | To understand the extent of adoption and use of digital health tools and to identify key perceived psychological motivators of technology use. | 92 | Mixed method study | N/A | digital health technologies need to be further explored to design better tools and interventions that address this population's interests and enhance the competencies to manage self and child health. | | Relied on self-reports, potential for bias |
| Guo et al. (2018), USA | To survey API (Asian/ pacific islander) women about their experience with health literacy and understanding medical terminology | 291 | Qualitative study | N/A | Demonstrated the need for increased health literacy | Improving patient provider interactions, limiting misunderstanding, and utilising innovative technology | Looked at health literacy and not English proficiency, which is potentially a barrier |
| Matta Machado Fernandes et al. (2021), Brazil | To analyse associations between social-demographic characteristics, childbirth information, perceived knowledge regarding the EBP, Normal Birth and Caesarean and women's behaviour. | 555 | Mixed methods study | N/A | Identifying the perceived barriers and knowledge towards EBP, understanding the high rate of C/S and implementing plans to increase EBP and Doula use | Increasing women's knowledge is part of the path to promote a positive childbirth experience. Gave the women who participated a chance to critically reflect upon Brazil's maternal health care scenario and advocate for their choices, desires, and rights. | Limitations inherent to a post-intervention cross-sectional design, where participants answered about before and after knowledge and childbirth experiences at a single point in time, without a non-exposed comparison group. |
| Okesene-Gafa et al. (2016), New Zealand | To assess pregnant women in an ethnically diverse areas to understand and observe differences in their thoughts and beliefs regarding gestational weight gain and the potential adverse health effects. | 422 | Qualitative study | N/A | Understanding that ethnic minority groups like the Māori have different accessibility to healthy food/knowledge about the importance of healthy food | Potential for women to participate in other nutritional studies | Addresses were not collected, and calculation of deprivation index was not possible. No follow up was done |
| Peters et al. (2017), The Netherlands | To evaluate the effect of a culturally competent educational film (CCEF) on informed decision making (IDM) regarding prenatal screening (PS) in a study population consisting of multicultural pregnant women | 373 | Cross sectional study | Culturally competent educational films (CCEF)- Digital Health Intervention | Found that Culturally Competent Educational Film (CCEF's) as part of the counselling on Prenatal Screening (PS) do lead to an increase in knowledge and Informed Decision Making (IDM) in specific groups. | Recommend an update of the CCEF's to the current context of PS (including the Non-Invasive Prenatal Test, NIPT). A randomized controlled trial would be useful to further build on the evidence for the films | Short inclusion period of the intervention group women, potential for bias |
| Petersen et al. (2021), Denmark | To assess potential ethnic differences in women's reported knowledge about how to manage symptoms of pregnancy complications | 1899 | Cross sectional study | MAMAACT training- Community based intervention | This study shows that immigrant women reported lower certainty in how to manage symptoms of pregnancy complications compared to women of Danish origin. | Efforts are needed to improve communication about pregnancy complications within the maternity care system. | Ethnicity definition is complex and varying depending on context. Unequal distribution between ethnic minorities |

*(Continued)*

**Table 2.** (Continued)

| Author, Date and country | Aims/ Objectives | Sample size | Study Design | Intervention (if applicable) | Outcome(s) | Findings and Recommendations | Considerations |
|---|---|---|---|---|---|---|---|
| Smart et al. (2021), UK | To measure the awareness of Ectopic Pregnancy signs and symptoms in women in an East London population | 400 | Cross-sectional cohort study | N/A | Increased researcher awareness about the social determinants of health especially in pregnancy | 1. Develop a targeted plan of disseminating patient information effectively to the vulnerable sections of the population with better designed patient leaflets in multiple languages and information points. 2. Launch a series of education meetings and groups where women attend for further information. 3. increased media and celebrity involvement | More follow-up care to understand other aspects of pregnancy and health literacy in ethnic minorities in London |
| Yee et al. (2020), USA | To use SweetMama for 2 weeks and provided qualitative (interviews) and quantitative (validated usability/ satisfaction measures) feedback on the SweetMama experience. | 22 | Qualitative study | SweetMama phone app- Digital intervention | Saw an increased use of the app, but health literacy increased was not measured | Opportunity to follow up and measure the effectiveness on the app for health literacy. | Limited data used for follow up and small sample size were challenges |
| Yee et al. (2022), USA | The objective was to evaluate user experiences with text message–based support intervention called Texting for Diabetes Success (TDS), including feasibility, acceptability, and areas for improvement. | 26 | Qualitative study | Texting for Diabetes Success (TDS) application- Digital Intervention | Enhanced motivation and improved knowledge and comfort with diabetes self-care activities. | The next steps include enhancing the interactive features of TDS, scaling to a high-tech mHealth app, and investigating the effect of TDS on maternal and perinatal outcomes. | Findings are not widely generalizable. Future work may investigate differences in support needs and mHealth adoption based on diabetes type, prior experience with gestational diabetes, or other demographic characteristics, such as educational attainment |

Jacoby et al., solely observed Somali Immigrants to Maine [26]. Another specific example of a BAME population came from the study from Thailand by Gilder et al [12] health literacy among ethnic minority population of Myanmar women. Two of the observed looked at indigenous populations; Okesene-Gafa et al observed Māori women, and Guo et al. observed Asian and Pacific Islander (API) women [27, 28]. A few of the studies only mentioned ethnic minority women or non-western immigrants in general to be compared to white women [24, 29, 30].

The studies were divided into two distinct groups: those measuring the impact of health literacy on BAME groups (47%) (n = 9) and those implementing interventions that aimed to enhance health literacy in BAME pregnant women (53%) (n = 10). This is an important distinction to make when looking at the impact that the studies made based on their aims and objectives.

Health literacy was measured in a variety of ways. In five studies (56%), the level of health literacy was based on information gathered from focus groups or interviews, as illustrated by Barnett et al., that captured Black women's childbirth experiences and how their lack of health literacy adversely affected their experience [31]. Additionally, surveys and questionnaires were

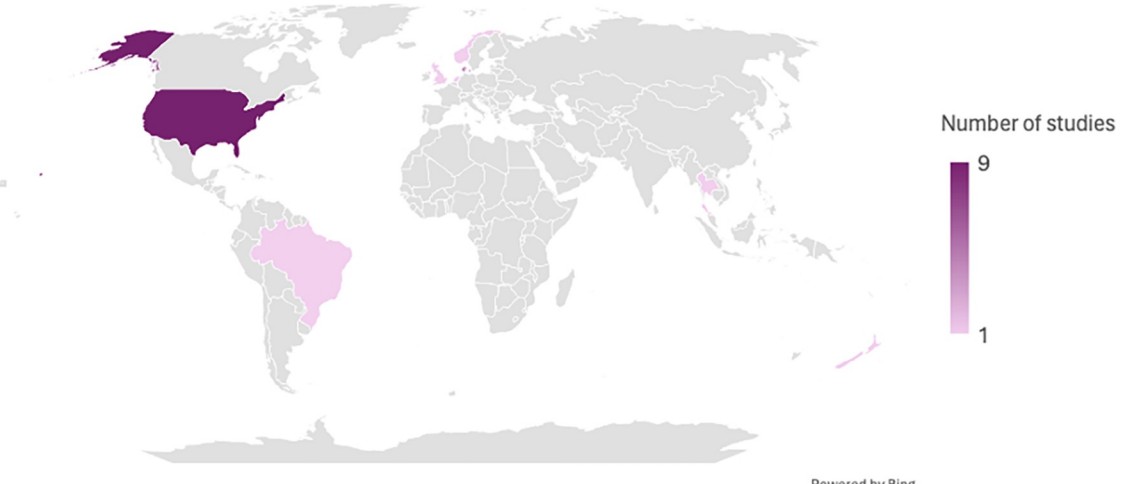

**Fig 2. Number of the included studies per country in this review.**

also popular tools encompassing 44% of the studies (n = 4). For example, Okesene-Gafa et al., used a nutritional health literacy survey [28, 30].

Out of the ten studies that used an intervention method, five (50%) used digital health interventions. Different types of digital interventions were used throughout the studies. Notably, 60% (n = 3) of these digital interventions used a texting service [29, 32, 33], 20% (n = 1)used mobile phone applications [32], and another 20% (n = 2) used educational films [34]. Community-based interventions were also prominent, with three (60%) of the five studies utilising a specific type of consortium or questionnaire as interventions. For example, the Newest Vital Sign (NVS) questionnaire and focus group were used in Jacoby et al., to measure and improve the health literacy of Somali immigrants in Maine, US [26]. Similarly, a consortium was designed was designed in Alio et al for Black women to decrease maternal mortality [35]. Two studies (20%) had a unique community training (MAMAACT) for patients and healthcare providers. The intervention incorporated patient awareness of pregnancy through leaflets, informational guides, and midwife and nurse training to help support ethnic minority groups. It was founded in Denmark to reduce social disparity in maternal mortality and early infant death.

## Synthesis of results

The impact of the studies consistently highlights the importance of health literacy during pregnancy as well as the need for greater self-advocacy in hospital settings. This is particularly evident in the study by Barnett et al., which focuses on the experience of Black women giving birth in the US. The study emphasised the importance of advocating for oneself as an ethnic minority, especially with providers who are not from ethnic minorities [31].

Of the nine studies that measure health literacy's impact on health outcomes and patient experience, 33% (n = 3) found suboptimal health literacy among BAME pregnant women, indicating the need for further research. Recommendation from studies, such as Barnett et al., specifically recommend a strategic plan to help Black women discuss their care with providers, with the goal of both increasing health literacy and patient experience [31]. While studies such as Okesene-Gafa et al., discussed how nutritional health literacy could be implemented in studies outside of pregnancy and Gilder et al., discussed how the result of the study gives important

information for how to pilot other public health campaigns that require basic health literacy [12, 28]. Although the studies generally advocate for increased health literacy in BAME groups, only one study by de Matta Machado Fernandes et al., described the impact that EBP and knowledge of healthcare practices had on health outcomes [11]. More commonly, studies focused on the impact health literacy had on patient experience rather than the impact of health outcomes of pregnant women. [31, 35].

Among the 10 studies employing interventions to improve health literacy in BAME pregnant women, four of the five (80%) digital interventions showed an increase in health knowledge and understanding, particularly within BAME groups. While they demonstrated an increase of knowledge, there was minimal impact on the maternal health outcomes or decreasing rates of maternal mortality. The most prominent were studies using text-messaging programs like T4B, studied by Blackwell et al., which showed a positive impact on increasing health literacy of ethnic minority women in Brooklyn, New York [29]. The one study that utilised a phone application, by Yee et al., only measured an increased use of the app, not an increase in health literacy [32]. Community-health interventions showed varied outcomes. Two of the studies (40%) showed an increased understanding of the importance between the patient/provider relationship to understanding health highlighting the importance of patient experience [35, 36]. While both MAMAACT training-based interventions demonstrated that BAME groups have decreased health literacy, only one of them, by Petersen et al. [25] showed a significant increase in health literacy due to the intervention.

## Comment

### Principle findings

This review aimed to evaluate the impact of health literacy on maternal health outcomes and patient experience among BAME pregnant women. Findings from 19 articles demonstrated lower baseline health literacy in BAME women than in non-BAME women [25, 26, 30, 36, 37]. While results varied, the majority of digital interventions worked well in increasing health literacy [29, 32, 34]. Overall, there were two pathways in looking at health literacy of BAME pregnant women: a) through studies that measured health literacy, and b) through studies that implemented interventions to increase health literacy. Measurement studies used surveys, questionnaires, and interviews offering insights into health literacy of BAME pregnant women and patient experience [20, 22, 34]. Intervention, whether community-based (consortiums, listening sessions, and group discussions) or digital interventions (like phone applications and text messaging services), demonstrated improved health literacy for BAME pregnant women [29, 32, 34]. While the studies that measured health literacy provided instruction on how they could be implemented in different settings. It is important to note that though the interventions found throughout the studies demonstrated an increase in health literacy within BAME populations, maternal health outcomes were minimally impacted. Overall, the 19 studies show a growing research interest in this area, especially in the last five years, there is still much more research needed in this area. The scoping review highlights the need for evidence-base practice in health literacy assessment through the various measurement approaches used across studies.

Digital health interventions, including user driven applications, text messaging systems, and *eHealth* websites, are becoming more popular in providing healthcare as well as aiding in increasing health literacy [38]. This shift to the digital realm creates a unique research opportunity, particularly in the domain of digital health literacy, especially within BAME communities. Van Kessel et al. proposes a four level framework for digital health literacy: functional, communicative, critical, and translational [39]. The phenomenon of the *digital divide, outlined by*

Scheerder et al., highlights disparities in access to digital information technologies, mirroring the social determinants of health [40, 41]. Equal access to digital health literacy is affected by variables such as race/ethnicity, geographic location, and socioeconomic status [39, 40]. The digital literacy divide widens the gap in health literacy, with a need for future research to demonstrate how digital health literacy can be equitable.

## Comparison with existing literature

In health literacy research, a gap exists in translating findings into implementation for specific populations [42]. To bridge this gap, conceptual frameworks illustrating the effect of health literacy as well as providing practical results could be implemented into health systems, particularly for BAME groups.

For example, a study by Fernandez-Gutierrez et al. compiled a systematic review of health literacy frameworks aimed to help immigrants [42]. The interventions and themes were listed, but there was a need for more discussion on results and the communities that were impacted. There is opportunity for future studies that look at the impact of these frameworks in a randomised controlled setting, moving beyond creating conceptual frameworks to observing how they affect the groups they are designed to help. Particularly with digital health interventions, being able to test them is pivotal to observing how they work. Rhodes et al., carried out a systematic review of digital health literacy tools for pregnancy and only included RCTs and pilot studies to demonstrate the importance of being able to use the devices [43].

## Strengths and limitations

There are many strengths and limitations in this review. In exploring the challenges faced by the studies, differences in measuring community-based and digital interventions appear. While digital technology enables user engagement, monitoring applications and texting program signups, community-based interventions have different and non-universal methods of measuring data and health outcomes. This difference is a potential limitation making interventions harder to compare to one another in terms of their potential impact.

Other challenges varied based on country or BAME target population. Fragmented social care, accessibility to technology, and survey and reporter bias were all mentioned as limitations throughout the 19 studies. By not identifying a specific BAME population, ethnic minority challenges were grouped together in a way that could cause potential bias.

While this review has limitations, it also has strengths. A key strength is the active engagement of stakeholders, particularly the Patient and Public Involvement and Engagement (PPIE) group, throughout the entire process on how best to utilise the findings of the review and discuss the impact they have. This co-production with the PPIE group was able to give the research validity as well as inspire the authors for future research questions.

## Conclusions

BAME women suffer from higher rates of maternal mortality and have decreased levels of health literacy. Health inequalities that persist in BAME groups need to be mitigated through more research and investment into interventions. The impact of digital and community-based public health interventions was explored and found to be successful tools in increasing health literacy in BAME women.

The global aspect of this review identified different interventions that aim to help BAME pregnant women that could be explored in other settings and provided recommendations for future research. Most importantly, creating a more patient-focused, diverse, and inclusive

approach to prenatal care creates more opportunities for interventions to be developed to increase health literacy and patient experience.

## Implications

The maternal mortality (MM) rate as of 2021 within the UK is over 11/100,000, with BAME women and women in deprived communities dying two times more frequently than affluent white women [44]. Most of the studies in this review were from Denmark and the United States, so it is important to compare their MM rates. Denmark has a MM of 5/100,000, while the US has over 32/100,000 [45, 46]. The difference between these rates contributes to the need for the study as well as the consistent fact that BAME women are dying at higher rates than white women.

Studies have proven that evidence-based practice (EBP) and community-based interventions can help in reducing maternal mortality rates [47, 48]. Community-based interventions, like focus groups, consortiums, and the MAMACAT training showed increased health literacy rates among BAME women.

Similarly, digital health interventions that focus on maternal health and wellbeing, especially for pre-existing conditions like diabetes and Human Immunodeficiency Virus (HIV) would be extremely beneficial. This is investigated by Zunza et al., [49], in which a texting program is implemented for breastfeeding women with HIV in South Africa. Ensuring universal access to digital health interventions is essential, seeing the global dichotomy between who has access to internet and digital technology and who does not.

To achieve the SDGs, there is an urgent need to provide further funds to non-governmental organisations (NGOs), charities, and community groups for both digital and community focused prenatal health interventions. Pallas et al.'s work on the sustainability of community healthcare workers in low and middle income countries [50] has found their role to be crucial to deliver and implement the interventions evaluated in this scoping review in BAME communities. Investment and subsequent scale up of the role of community healthcare workers should be a top priority to increase the health of pregnant women in BAME communities.

## Supporting information

**S1 Checklist. This is the PRISMA checklist.**
(DOCX)

**S1 Table. This is the raw data implement in Table 2.**
(XLSX)

**S1 Appendix. This is full the search strategy used.**
(DOCX)

## Acknowledgments

**Disclaimer:** This independent research is supported by the National Institute for Health and Care Research Applied Research Collaboration Northwest London and funded by the Imperial College London Library services.

## Author Contributions

**Conceptualization:** Sarah E. Feldman, Laura Lennox, Keivan Armani.

**Data curation:** Sarah E. Feldman, Laura Lennox, Natasha Dsouza.

**Formal analysis:** Sarah E. Feldman, Laura Lennox, Keivan Armani.

**Investigation:** Sarah E. Feldman.

**Methodology:** Sarah E. Feldman, Laura Lennox, Keivan Armani.

**Project administration:** Sarah E. Feldman.

**Resources:** Sarah E. Feldman, Laura Lennox, Natasha Dsouza, Keivan Armani.

**Software:** Sarah E. Feldman, Keivan Armani.

**Supervision:** Laura Lennox, Keivan Armani.

**Validation:** Sarah E. Feldman, Laura Lennox, Natasha Dsouza.

**Visualization:** Sarah E. Feldman, Natasha Dsouza.

**Writing – original draft:** Sarah E. Feldman.

**Writing – review & editing:** Laura Lennox, Natasha Dsouza, Keivan Armani.

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
