## [Decision Letter · Decision Letter 0]

2 Aug 2024

PONE-D-24-25675Exploring the Impact of Health Literacy on Pregnant Women from ethnic minority groups A Scoping ReviewPLOS ONE

Dear Dr. Armani,

Thank you for submitting your manuscript to PLOS ONE. After careful consideration, we feel that it has merit but does not fully meet PLOS ONE’s publication criteria as it currently stands. Therefore, we invite you to submit a revised version of the manuscript that addresses the points raised during the review process.

**Thank you for your interesting paper.   Please take on board all the comments below from the two reviewers.   Also please take on board the following:** **1) With the patient representations did you use PCC or PEO or PICO to design the research question?****2) With your narrative analysis - did you use thematic analysis - how did you do your analysis of text?****3) In the PRISMA flowchart - the word 'wrong' is used - can you use an alternative word for why excluded rather than wrong. **

We look forward to receiving your revised manuscript.

Kind regards,

Julia Morgan

Academic Editor

PLOS ONE

Journal Requirements:

2. Please include a copy of Table 3 which you refer to in your text on page 7.

Reviewers' comments:

Reviewer's Responses to Questions

**Comments to the Author**

1. Is the manuscript technically sound, and do the data support the conclusions?

Reviewer #1: Partly

Reviewer #2: Yes

2. Has the statistical analysis been performed appropriately and rigorously? 

Reviewer #1: N/A

Reviewer #2: N/A

3. Have the authors made all data underlying the findings in their manuscript fully available?

Reviewer #1: Yes

Reviewer #2: Yes

4. Is the manuscript presented in an intelligible fashion and written in standard English?

Reviewer #1: Yes

Reviewer #2: Yes

5. Review Comments to the Author

**Reviewer #1:** Thank you for undergoing this critical research on "Exploring the Impact of Health Literacy on Pregnant Women from ethnic minority groups". This work highlights the need for more primary research that investigates the impact of health literacy on maternal outcomes and patient experience among these marginalised groups. Very well done to all the authors!

There are a few sections that require minor revisions to improve upon the current manuscript:

1. The setting was not very clear to the reader.

Issue (I): The included articles and conclusion suggest a global focus while some sections of the introduction point towards the United Kingdom and the United States. For example, Page 3 paragraph 1 suggests a focus on the UK while Page 3 paragraph 3 suggests an focus on the US and the UK. This is not the case with Page 3 paragraph 3, where several articles from different countries have been included, majority of them been from Denmark and the US.

Suggestion (S): You may want to establish the setting more clearly and introduce data from more settings in the introduction rather than the UK or the US as they do not seem sufficient, considering the global focus. Alternatively, you may consider highlighting the lack of data from other places, if this is the case.

2. Conflicting date limitations in eligibility criteria.

I:On page 5 paragraph 2, the statement "we checked other systematic reviews to see similarities and differences in already published works" is suggesting that this is a systematic rather than scoping review.

S:You may want to remove "other".

3. Conflicting timelines for data retrieval.

I:On page 5 paragraph 3, you state that "Studies published before 2013 were excluded". However, you also state that "The data collected and published in the report spans from 2011-2021 and is the most recent and up to date evidence of the impact of race within the healthcare system".

S: Please clarify the correct date filter used.

4. Crosscheck the content of Table 1.

I: On page 5 paragraph 2, you write that "Terms involving pregnancy AND health literacy AND BAME or ethnic minority group were combined for maximum results, see Table 1 for detailed search terms that were used" but on page 6 paragraph 1, you write that "Studies excluded during the full text screening stage were put into the categories described in Table 1". However, Table 1 is labelled "Table 1: Inclusion and Exclusion Criteria".

S: Please correct and label tables accordingly. Include the table on search terms.

5. It is not clear how the conclusion that maternal health outcomes were not impacted by health literacy for BAME groups was made.

I: On page 10 paragraph 2 you state that "Although the studies generally advocate for increased health literacy in BAME groups, only one study by de Matta Machado Fernandes et al., described the impact that EBP and knowledge of healthcare had on health outcomes (18). More commonly, studies focused on the impact health literacy had on patient experience rather than health outcomes (28,32)". Also, on page 11 paragraph 1, you state "It is important to note that though the interventions found throughout the studies demonstrated an increase in health literacy within BAME populations, maternal health outcomes were not impacted".

S: Please check and correct for clarity.

6. Present the presence or absence of key findings on the aim/objectives

I: On page paragraph, you reiterate that "This review aimed to evaluate the impact of health literacy on maternal health outcomes and patient experience among BAME pregnant women". However, these findings are not clearly presented.

S: Clearly identify the impact (or lack of impact or data) of health literacy. Were patient experiences and outcomes better or worse or not impacted or not reported in relation to the health literacy of the women?

Once again, it was a pleasure to review this outstanding piece of work. Thank you.

**Reviewer #2:** Thank you for submitting this manuscript. This is an important topic to address. However, I noticed that the journal's criteria only considered systematic reviews...please ignore if the journal have been consulted about this. Some additional comments are provided below.

Abstract

• Main research question and key findings are summarised well in the abstract.

Introduction

• Would be good to have some justification of why there is a focus on adverse health outcomes in pregnancy.

• First paragraph of page three – last sentence – “The concept of health literacy is directly linked to the social determinant of health and thus connected to socioeconomic status (6,7).” Should socioeconomic status be first?

• Second paragraph of page three – first sentence – why does maternal have a capital M?

• Third paragraph of page three and second sentence – et al. is in italics.

• Best to replace Lang (reference 12) with a more recent reference if available or not include.

• Good use of background info to justify research question.

Results

• Page 7 – last line “Articles were only selected if they were published after 2013.” – can be removed as it is part of Methods.

• Page 8 – first line. Dash not needed.

• Page 9 – penultimate sentence of third paragraph: “The intervention patient incorporated patient awareness of pregnancy through leaflets,…” Was the first use of “patient” used in error?

• Page 12 – first sentence of third paragraph doesn’t make sense: “For example, a study by Fernandez-Gutierrez et al. compiled a systematic review of health literacy frameworks aimed to help immigrants (40) interventions and themes yet needing more discussion on results.”

• There is reference to a table 3 but this does not appear to be included – should this refer to table 2?

6. PLOS authors have the option to publish the peer review history of their article (what does this mean?). If published, this will include your full peer review and any attached files.

Reviewer #1: No

Reviewer #2: **Yes: **Sereena Raju

---

## [Author Response · Author response to Decision Letter 0]

17 Sep 2024

We have responded to all of the comments by the jounral (Editor) and the two reviewers.

---

## [Editor Report · Decision Letter 1]

23 Sep 2024

PONE-D-24-25675R1Exploring the Impact of Health Literacy on Pregnant Women from ethnic minority groups A Scoping ReviewPLOS ONE

Dear Dr. Armani,

Thank you for submitting your manuscript to PLOS ONE. After careful consideration, we feel that it has merit but does not fully meet PLOS ONE’s publication criteria as it currently stands. Therefore, we invite you to submit a revised version of the manuscript that addresses the points raised during the review process.

**Thank you for making the corrections.   There is a reference missing for global south in the introduction - the words (ref) are present - please correct and resubmit.   Please check through the paper for any other typos.**

We look forward to receiving your revised manuscript.

Kind regards,

Julia Morgan

Academic Editor

PLOS ONE

Journal Requirements:

Additional Editor Comments:

Thank you for making the corrections. There is a reference missing for global south in the introduction - the words (ref) are present - please correct and resubmit. Please check through the paper for any other typos.

---

## [Author Response · Author response to Decision Letter 1]

4 Oct 2024

We have removed the word ref and have added five references for Global South statement. 

Regards,

Keivan Armani

---

## [Editor Report · Decision Letter 2]

9 Oct 2024

Exploring the Impact of Health Literacy on Pregnant Women from ethnic minority groups A Scoping Review

PONE-D-24-25675R2

Dear Dr. Armani,

We’re pleased to inform you that your manuscript has been judged scientifically suitable for publication and will be formally accepted for publication once it meets all outstanding technical requirements.

Kind regards,

Julia Morgan

Academic Editor

PLOS ONE
---

## [Editor Report · Acceptance letter]

16 Oct 2024

PONE-D-24-25675R2 

PLOS ONE

Dear Dr. Armani, 

I'm pleased to inform you that your manuscript has been deemed suitable for publication in PLOS ONE. Congratulations! Your manuscript is now being handed over to our production team.

Kind regards, 

on behalf of

Dr. Julia Morgan 

Academic Editor

PLOS ONE